# The Science and Social Validity of Companion Animal Welfare: Functionally Defined Parameters in a Multidisciplinary Field

**DOI:** 10.3390/ani13111850

**Published:** 2023-06-01

**Authors:** Lauren I. Novack, Lauren Schnell-Peskin, Erica Feuerbacher, Eduardo J. Fernandez

**Affiliations:** 1Department of Special Education, Hunter College, New York, NY 10022, USA; ls2875@hunter.cuny.edu; 2Department of Animal and Poultry Sciences, College of Agriculture and Life Sciences, Virginia Tech, Blacksburg, VA 24060, USA; enf007@vt.edu; 3School of Animal and Veterinary Sciences, University of Adelaide, Adelaide, SA 5005, Australia; edjfern@gmail.com

**Keywords:** companion animal, dog, behavior analysis, welfare, social validity, animal training, behavior intervention

## Abstract

**Simple Summary:**

Pet dogs are more prone to exhibit challenging behaviors than ever before. Dog trainers are increasingly tasked with helping pet owners resolve behavior issues, not just teach their charges good manners. The interventions used by professionals to help ameliorate behavior complaints must be evidence-based and include the effectiveness of the intervention, how the intervention is perceived by the learner, and how the intervention affects the learner’s quality of life before, during, and after behavior intervention procedures. The objective of this paper is to review literature from multiple scientific disciplines and demonstrate how concepts from applied behavior analysis and the animal welfare sciences can be used together to ensure that the animal undergoing intervention experiences good welfare during the training process.

**Abstract:**

Social validity refers to the social significance and acceptability of intervention goals, procedures, and outcomes. Animal practitioners, who are often guided by the principles of ABA, lack the benefit of verbal participants (at least with respect to target animals) with which to assess a client’s needs and preferences. The study of a learner’s welfare is useful for determining areas where intervention is needed or how the learner feels about an intervention that is underway. Three tenets of animal welfare measurement include physiological function, naturalistic behavior, and affect, where affect refers to private events, including emotions, which are a function of the same variables and contingencies responsible for controlling public behavior. The development of new technologies allows us to look “under the skin” and account for subjective experiences that can now be observed objectively. We introduce the reader to tools available from the animal welfare sciences for the objective measurement of social validity from the learner’s perspective.

## 1. Introduction

Scientist practitioners have an ethical responsibility to possess a comprehensive understanding of behavior, without which behavior analysis principles may be inadequately applied. It is also the duty of scientist practitioners to understand how other behavioral sciences contribute to the application of behavior analysis, educate the public about a learner’s needs in order to change public opinion and action [1], and disseminate knowledge about interventions that are both effective and ethical. Animal practitioners, who are often guided by the principles of applied behavior analysis (ABA), lack the benefit of verbal participants (at least with respect to target animals) with which to assess a client’s needs and preferences. To do so, animal practitioners (trainers, behavior consultants, and animal behaviorists) incorporate tools developed by animal welfare scientists [2,3] to assess how animal’s feel about their own circumstances and overall quality of life (QOL). 

Research about the domestic dog provides insight into their cognition, including but not limited to their sensitivity to, perception of, and relationships with humans, interpretation of human vocal and physical cues, discrimination learning, executive functioning, spatial and visual processing, memory, sense perception, umwelt, and the way cognition changes throughout development and aging [4,5,6,7,8,9,10,11,12,13,14,15]. This research is critical for our understanding of the salience of different stimuli to the domestic dog [16,17], without which it would be impossible to design living environments in which they can succeed and thrive, nor learning environments that they find reinforcing. 

When designed from a multidisciplinary perspective including cognition, ethology, welfare, and behavior analysis, interventions can modify interactions between animals and their conspecifics or humans [18,19,20,21], resolve unwanted behaviors [22,23,24,25,26,27,28,29,30,31,32,33,34,35,36,37], temporarily increase enrichment usage, and expand the learner’s behavioral repertoire [38] while ensuring good welfare. When an intervention’s scope does not consider how the intervention impacts the learner, interventions focus on the elimination of behavior via punishment and aversive control [39], leading to outcomes that include fear, anxiety, pain, stress, aggression, and a negatively impacted dog–handler relationship [18,32,34,40,41,42,43,44,45]. Effectiveness is important, especially as it pertains to social validity [46,47,48]. However, the learner’s experience of the intervention matters, particularly as it relates to their choice to participate in any procedure [49,50].

Until recently, the exploration of canine behavior interventions relied on group design rather than the robust technologies of behavior change developed by behavior analysts [51,52]. Much of our understanding of training success comes from owner questionnaires, surveys, or between subject studies that focus on measures of central tendency (means, medians) rather than individual differences, and often do not meet best practices for skill acquisition and behavior change, or fail to report on metrics including operational definitions, criteria setting, generalization, maintenance, treatment integrity, and interobserver agreement. Many interventions are under-developed and fail to report adequately on what was carried out and what components contributed to the intervention’s success or failure; much of the animal intervention literature is therefore evidence-inspired rather than evidence-based [53]. For a review of the common ways behavior interventions are designed, implemented, and reported in the animal literature, see [53] (Table A2).

Over the past 10 years, functional analysis (FA) was used to examine the variables responsible for various problem behaviors in companion dogs prior to the implementation of function-based treatments. Analyses and their function based interventions successfully treated jumping up behavior [23,31], stereotypic behavior [29], resource guarding [27], kennel aggression and leash pulling in a shelter environment [37], demonstrated that access to the owner functions as a reinforcer for pet dogs [54], and showed that owners are capable of implementing FA and treatment of mouthing behavior [35]. Others demonstrated the success of interventions grounded in behavior analysis, using baseline measures to demonstrate treatment effectiveness [22,24,26,28,32,55]. As the body of literature pertaining to dog welfare and behavior interventions increases, so does the commitment to honoring the human–animal bond. It is easier to encourage the use of interventions that are both function and welfare focused when those interventions improve the lives of both animals and the humans who care for them. Yet the measurement of an animal’s welfare before, during, and after intervention, which is necessary to ensure that an intervention is having only a positive effect on the learner, is yet to become standard practice. 

The objective of this paper is multifold: (1) to introduce the applied behavior analysis community to tools available from the animal welfare sciences for the objective measurement of affect, (2) to illustrate the importance of programming for true choice to determine how an animal feels about an intervention that is underway, (3) to propose that measurement of a learner’s welfare be incorporated into data collection and analysis before, during, and after behavior intervention, and (4) to encourage multidisciplinary collaboration.

## 2. Animal Welfare

Welfare describes the state of an animal at a given point in time. The study of a learner’s welfare is useful for determining areas where intervention is needed or how the learner feels about an intervention that is underway. In this vein, the “Five Freedoms” model [56,57,58] was introduced to alleviate suffering experienced by captive farm animals and provided an early framework for identifying stimuli that negatively impact animal welfare and cause negative affective states. Affective states are subjective states that are experienced as pleasant or unpleasant rather than hedonically neutral [58]. In the “Five Freedoms” model, welfare was measured by individual animals’ freedom from experiences such as thirst, hunger, discomfort, pain, injury, disease, fear, and distress, and the freedom to express “normal” behavior (i.e., species-typical or otherwise individually desired responses) [56,59]. This preliminary model labeled welfare “good” if the animal was not suffering and had the ability to cope with their environment. This early foray into animal welfare led to interventions that helped move the needle from poor to improved welfare states and paralleled the concurrent behavior analytic shift toward focusing on what skills a learner needs to acquire rather than which responses need to be eliminated [60]. 

The “Five Domains” model, an extension of the “Five Freedoms” model, was developed to assess the compromised welfare of animals used in research, teaching, and testing [61]. The “Five Domains” model outlines five areas of potential welfare compromise (i.e., nutrition, environment, health, behavior, and the mental states arising from these factors). This framework is consistently updated, with measures of positive welfare and the ability to thrive now included [62,63,64,65]. Most recently, Mellor et al. [2] amended the domain ‘behavior’ to ‘behavioral interactions’, expanding the domain to include behaviors that contact not only the environment and conspecifics, but also humans themselves. 

Domains 1–3 of the “Five Domains” model allow for the systematic identification of external circumstances, and the internal physical states and affects associated with them, of survival–critical behaviors related to nutrition, the environment, and health. Domain 4 allows for the systematic identification of situation-related behaviors and the effects associated with them, and highlights the importance of agency. For example, barren environments, inescapable sensory impositions, and choice restraints are labeled as welfare compromising, where free movement, exploration, and foraging are labeled as welfare enhancing. Domain 5 allows for the systematic identification of a learner’s mental state as it is brought about by the previous four domains; constraints on an environment-focused activity are associated with depression, where congenial sensory inputs are associated with gratification. This framework mirrors the assertion made by radical behaviorists that responses, including private events, are caused by the environment. Each domain is to be assessed in relation to the animal’s “species-specific behavior, biology, and ecology considered in relation to their specific physical, biotic, and social environment” [66]. The model can be used to identify a wide range of welfare states, grade welfare compromise and enhancement, enable and monitor interventions when corrective action is required, engender empathy towards learners through the identification of the wide range of positive and negative welfare-relevant experiences that can be identified, and facilitate the consideration of quality of life when the model is used repeatedly over time. The “Five Domains” were used for animal welfare management, particularly in zoo settings, and there is no reason why this model could not be incorporated into any setting in which a learner is under the care and control of others. This is especially true when the learners’ behavior is being targeted for intervention. Just as human beings “accept no limits to their own wellness” and “cannot be too well” [67], setting the events for optimal welfare should be the goal of any individual responsible for the care of any learner. 

Animal welfare is an increasingly public subject, with the perception of an animal’s welfare used by the public to determine whether a procedure or system is acceptable. The same environments and responses may be deemed acceptable in one scenario or unacceptable and targeted for modification and intervention in another. Ethics guides decisions about what others feel is tolerable for the organism. This can be in alignment or at odds with what the organism values and needs, with welfare compromise likely when the values of the organism and others are at odds. For example, beagles are known for their tenacity for following a scent. When used for a human’s purpose (i.e., sniffing out bed bugs, bombs, or drugs), sniffing behaviors are valued. When a pet owner is attempting to walk their beagle on a leash, incessant pulling is not valued and may lead to frustration and frustration-related human behaviors, such as yelling or yanking on the leash. Where social validity historically placed the values and concerns of others above the learner and was therefore measured subjectively, welfare is both evidence and values based [68]. The lens through which welfare is viewed, and by which welfare measuring technologies are developed, are also influenced by an individual’s philosophy and morality, which are shaped by the mores of their society, tribe, family, and personhood. This highlights the importance of ensuring that welfare is measured objectively, with what the learner values included within the measurement tool. 

While the movement toward “freedom to” and away from “freedom from” can be seen in the recent animal welfare literature [3,65,65,66], veterinarians, who are the primary source of information for most pet owners, currently consider the ability to cope as the parameter by which to assess if a companion animal has “good” welfare (American Veterinary Medical Association, n.d.-a). Yet the ability to cope does not define or guarantee good welfare. Coping is defined as having control of mental and bodily stability [69] or having the ability to adapt, where adaptation is the use of regulatory systems, with their behavioral and physiological components, to help an individual cope with its environmental conditions [70]. The inability to cope indicates welfare compromise, and as such frames veterinarians to focus on suffering reduction, which is a noble goal that aligns with the pathological framework from which they work [71]. The inability to cope with the environment is associated with reduced behavior variability, reduced species-specific behaviors, and the presence of stereotyped or abnormal behaviors [51]. Failure to cope is reflected in reduced fitness measured by shorter life expectancy, increased intervals between breeding periods, and fewer offspring [70], which are not metrics that veterinarians can easily measure. Physiological measures of stress, including temperature, heart rate, and respiratory rate can be used alongside visual inspection of the pet’s body condition [72], pain scales [73,74], and body language, but veterinarians must rely on owners conveying their concerns about their pet’s behavior, as QOL assessments for clinical practice are used only during end-of-life care to guide euthanasia decisions [75,76,77]. These surveys ask pet owners to rate easily operationalized concepts, including hurt, hunger, hydration, hygiene, and mobility. They also ask pet owners to rate happiness and “more good days than bad”, which are subjective and difficult to quantify without more robust, operationalized definitions or the inclusion of more systematic metrics. While they provide a guide for tracking an animal’s decline from terminal medical diagnosis to euthanasia, they do not include baseline measures when the animal is healthy, nor robust positive welfare indicators, so are insufficient for identifying how the welfare of apparently healthy companion animals could be improved through interventions or for guiding decisions regarding behavioral euthanasia.

Outside of the veterinary field, updates to welfare assessments aim to assess the animal’s QOL, where QOL is “the emotions of the dog both positive and negative, physical fitness and health, and the ability and opportunity to perform natural behaviors” [3]. The aim of welfare assessments is to facilitate the creation of environments and interventions that promote a good QOL, or a life worth living [66]. Doane and Sarbeno [78] proposed a model for dog welfare based on the five domains model, excluding the first two domains, and adding a sixth domain to evaluate the QOL of the dog owner. The authors found it was possible to construct a reliable questionnaire using questions from three places; a modified Canine Behavioral Assessment and Research Questionnaire (C-BARQ) [78,79]; a validated tool used in applied clinical settings to assess the severity of problem behaviors in dogs, a dog QOL questionnaire [76], and an owner QOL questionnaire [80]. One hundred eighty-five questions examined the typical behavior of the dog, including behaviors associated with fear, anxiety, aggression, excitability, attention, attachment, touch/pain sensitivity, and trainability, in addition to questions about the owner’s emotions, social and physical quality of life and stress, and the happiness, physical functioning, hygiene, and mental status of the dog. This pilot study demonstrated that this questionnaire may be suitable for further development as a tool for dog welfare assessment, where scores could illustrate areas of decreased welfare for both the dog and their caretaker, and when repeated aid in the assessment of how a behavior intervention impacts stakeholders. However, in order to create a tool that is not only accurate, but also likely to be adopted, further consideration should be given to the number of questions asked and time needed to complete and review the questionnaire.

### 2.1. Welfare and Environment

Measuring response allocation allows us to see how much time is spent, or the percentage of time spent, performing a response. Response allocation changes when an environmental change results in increased or decreased time spent performing the response. The occurrence of an event (the environmental change) is an inducer, which is a stimulus that occasions an (induced) activity. Unlike a discriminative stimulus or an elicitor, a close temporal relation is not required between the inducer and its response. A phylogenetically important event (PIE) is an unconditioned inducer that directly affects survival and reproduction. Thus, a PIE induces PIE-related activity, which includes operant activity that is related to the PIE as a result of the contingency [81]. Fitness-reducing PIEs will induce defensive activities that remove or mitigate danger, where fitness-enhancing PIEs induce fitness-enhancing activities. Some examples include seeing a predator and hiding and seeing prey and hunting. Similarly, induction can help explain ‘instinctive drift’ (i.e., misbehavior), as observed by Keller and Marion Breland [82], where non-reinforced behavior interfered with operant learning. For example, the presence of food elicits behaviors that are not targeted, and are not reinforced by food, yet persist and “strengthen”. When picking up coins, raccoons engaged in food washing behavior, which was not reinforced and interfered with the goal of having the raccoons deposit coins in a piggy bank. Food is phylogenetically important in that it is critical for survival. Hence, food is called a “primary” reinforcer, and is also an inducer [81,83,84]. In this instance, the coins induced the raccoons to dip the item into the box and bring them back out as they would with food into water, before rubbing the items together. The problem persisted in such a manner that the raccoons allocated all their time to food washing, and Breland and Breland could not get the raccoons to drop the coins into the box. 

Induction is related to allocation through contingency, which links an environmental event to an increase or decrease in activity. Response allocation is related to welfare due to the importance of response variability to welfare, with enclosure use and behavioral diversity measurement used to assess how environmental variables impact an animal’s quality of life [64,85,86]. Welfare is compromised when an organism spends too much time engaging in one activity to the detriment of other behaviors that are necessary for survival and wellbeing. A few reasons this may occur in dogs includes environmental deprivation [66], unwillingness to exit the home or interact with humans or conspecifics due to fear of the environment or something in it [87], neophobia, which causes the animal to avoid a novel objects or environments, preventing exploration [88], and stereotypies, which are repetitive or sustained goal-focused behaviors that do not change much from one situation to another and which are not a normal part of the ethogram (i.e., behavioral inventory) of the organism within the given context [89]. Some stereotypies include tail spinning [90], flank sucking [91], and fly snapping [92], which can be caused by or cause medical issues [92,93]. Welfare is compromised when the percentage of time allocated to one behavior or class of behaviors prevents the organism from engaging sufficiently in rest, play, exploration, and social interaction. The identification of inducers that promote behavior that support welfare enhancement, and the elimination of inducers that promote behavior correlated with welfare compromise, including but not limited to behaviors related to fear, anxiety, and stereotypy, is paramount in order to design an environment that supports a good quality of life. Additionally, variables responsible for changes in response allocation in natural settings must be identified so they can be replicated in restricted settings, where applicable [94]. 

In captive settings that are naturally restricted, humans control all resources, which are commodities or opportunities to perform specific activities in order to obtain an objective [95]. Behavioral repertoires and time budgets vary based on circumstances including environmental conditions and resource distribution [95]. In this scenario, “income” is a variable (for example, time or energy) that limits the resource acquiring response. Behavioral needs are requirements that must be met to allow for the organism’s effective functioning [95]. To determine if a behavior is one that “needs” to be performed, welfare compromise must be observed after the learner is blocked from performing that behavior [96]. An analogous operant, such as button pressing, can be trained to test the “price” of a resource, where a response, such as the number of button presses performed, could be used to determine the price of some commodity, or what an individual is willing to “spend” on that resource [97,98]. For example, Feuerbacher and Wynne [54] used “depressing a large button light” and “pawing a backstop” as analogous operants for responses that typically lead to dog owner access; these responses were used in a functional analysis to establish that access to owners is functionally reinforcing. While they required the dogs to perform the response only once, if singular button pressing was put on extinction, the number of button depressions performed could be used to determine how valuable access to an owner is. If welfare compromise was observed after access to the owners was not granted, or if the button was removed, responses that lead to owner access could be labeled behavioral “needs”. Blocking a learner from performing a behavior via environmental arrangement or response extinction is only one way that humans, who are an unavoidable part of a companion animal’s environment, cause poor welfare. 

The behavioral needs of dogs that have been identified include running, resting, playing, exploring, and positive, consistent interactions with humans; the environment should be of adequate size and complexity for these purposes [99]. For companion animals, who were genetically modified by humans, human environments comprise dogs’ natural ecological niche, and dog–owner attachment is functionally analogous to human infant–mother attachment [100,101]. Dogs have been observed wagging their tails more when granted access to human contact compared to access to conspecifics [102], prefer petting over food when the petting is provided by their owners in unfamiliar contexts [103], play more with conspecifics when receiving owner attention [104], and can communicate with owners via “showing” behavior [12]. Dogs and humans have a mutualistic relationship, with dogs bred to serve human purposes including hunting, guarding, and herding [105]. Further, humans and dogs support each other in stressful situations [106]. An oxytocin-mediated positive loop is facilitated and modulated by dogs and their owners gazing at each other [107]. Dogs even have a muscle responsible for raising the inner eyebrow intensely, which increases paedomorphism and may trigger a nurturing response in humans [108]. This muscle is not found in wolves.

All living beings that underwent natural selection are adapted to environments that allow them to thrive by every metric: affective, health, and behavior [58]; but what about animals that exist due to artificial selection? Over the last 200 years, dog breeding goals favored form over function and the behavior and personality in each breed is now highly variable [109,110,111]. Pet owners may not think of their pets, who are part of the family, as being held in captivity. Nevertheless, pets are restricted to the environment and resources provided by their humans. The natural behaviors of the domestic dog are ones that exist within their habitat, the human environment; yet dogs have widely diverse phenotypes and breeds are highly differentiated to display a wide variety of behaviors best suited to a large range of human settings [111]. The captive condition that is the pet home may not allow for the evolved adaptations of companion animals to match the challenges of their current circumstances, causing poor welfare when natural behavior no longer leads to beneficial consequences. Where herding behavior may be valuable on a farm, it is not appreciated when directed towards humans in an apartment! While pet owners’ homes should elicit natural behaviors that produce reinforcement for the dogs that occupy them, many modern pet homes lack the ability to meet the needs of companion dogs without purposeful construction, enrichment, and intervention. 

Bamberger and Houpt [112] found that the number of dogs exhibiting unwanted behaviors including aggression increased between 1991 and 2001, and veterinary behaviorists report experiencing an increase in the severity and complexity of their average patient (Christensen, E., personal communication, 20 October 2022). Some unwanted pet behaviors become stuck in a “sick social cycle” [113] with their owners’ behaviors. Fear and anxiety are reported as increasingly common behavioral disorders in dogs [87,112], with a prevalence ranging from 26.2 to 44% [43,87,114,115]. Fear and anxiety disorders seriously compromise the welfare of dogs and may lead to chronic stress, relinquishment by the owner, and euthanasia [87,116,117,118,119,120,121,122]. In zoos, wildlife centers, safari parks, and sanctuary settings, habitats are designed for the animals specifically and staff must adhere to basic requirements. While the level to which this is conducted is outside the purview of this paper, it is of benefit to both the animal and to conservation, research, and education efforts that animals in captive settings have good welfare [123,124,125]. Human dwellings are not built with pets’ needs in mind, and no welfare requirements exist for companion animals. However, we have an ethical responsibility to provide captive animals, including our pets, with environments that promote a good quality of life [124].

Dogs that find themselves in shelters are afforded even less autonomy. When an environment does not intrinsically meet an animal’s needs, enrichment should be supplemented. Enrichment typically refers to “inputs” or manipulanda, which provides for species-typical needs by meeting adaptive relevance [126]. Enrichment is the contingency between a response and stimulus or event, where the enrichment produces an observable, measurably improved state of well-being for the animal [37]. Kiddie and Collins [127] developed a scoring system to assess how enrichment impacted the welfare of kenneled dogs in a rehoming center (i.e., shelter). The authors compiled an ethogram of behaviors associated with anticipation, play, and relaxation, which were further categorized by whether the responses were provoked or unprovoked by human interaction (approached, touched, engaged in play, and physical examination). Dogs were put into four groups, and sorted by length of stay (more or less than 30 days) and environment (enrichment vs. no enrichment). Enrichment included removing the dog from the kennel, encouraging the dog to make body contact, giving a massage, speaking to the dog in a soothing voice, grooming with a soft brush, and 5 min of clicker training. They found that enrichment increased welfare scores and the assessment itself had good content evidence and criterion validity, but poor internal consistency. Staff were trained to promote response evidence of construct validity; however, interobserver agreement (IOA) data were not taken. 

Training can be enriching for dogs by “affording learning opportunities, and learning is considered to be enriching” [38,128]. Training may also teach skills that afford access to additional reinforcers. For example, successfully training dogs to “come when called” allows them the freedom to be off leash to choose where to go, who to interact with, and when and where to sniff. With this skill, owners can call their dogs away from livestock, wildlife, or any dangerous or inappropriate situation and release them when it is safe to do so. In this scenario, the response “coming when called” could be considered a behavioral cusp [129], as it is a requisite skill for being safely off leash, affording access to new environments, new opportunities to engage in species-typical behaviors associated with “feeling good” [130], and contact with new reinforcers, including but not limited to access to variable scents. Additional examples of enrichment include food puzzles from which animals must manipulate an object to obtain food [131], chews and toys meant for gnawing and dissecting [132], sniffing, which induces positive judgment bias [133,134,135], and play with humans or conspecifics [136]. Enrichment was shown to positively impact behavior welfare indicators by increasing behavior variability and exploration and decreasing stress-related behaviors. The ability to perform natural behaviors that are afforded through enrichment may have a regulating effect on the dog’s nervous system, and may be a setting event for other behaviors that human stakeholders find valuable, such as resting calmly at home. 

Enrichment strategies are purported to be most effective when targeting the primary sensory abilities of the species concerned [137]. Dogs rely on olfaction as their primary sense [130], and olfactory enrichment modifies the behavior of both pet and shelter dogs [133,134]. After exposure to cloths scented with ginger, coconut, vanilla, and valerian, dogs displayed reduced levels of vocalization and movement, with coconut and ginger also causing an increase in sleeping behavior [133]. By contrast, dogs spent more time moving and barking upon exposure to peppermint and rosemary [138]. Preliminary, non-peer reviewed research demonstrated that when on a walk, sniffing behavior lowers dog heart rates independent of their walking activity, with sniffing intensity positively correlated with heart rate reduction [139]. With increased freedom came increased sniffing, as dogs would sniff 280% more on a long leash compared to a short leash, and 330% more off leash compared to on a short leash. Given that dogs allocated proportionately more time to sniffing in relation to their freedom, we can assume that the opportunity to sniff is of value to dogs; further research is warranted to determine if sniffing is one of a dog’s behavioral needs. 

### 2.2. Environment Complexity and Choice

A balance must be found between predictability, which has been shown to reduce anxiety [140,141], and the variability and novelty afforded by more diverse environments and variable stimuli that allow for animal-driven choices [126]. Increased habitat complexity, environmental enrichment, contact with appropriate social groups, and training sets the occasion for species-typical behaviors and increases behavioral diversity, which is one measure of an organism’s welfare [64]. A large repertoire of behaviors allows for an increased chance of survival, as well as multiple routes to access primary reinforcement and escape, which supports the idea that programming to expand response classes may improve quality of life. Animals who are stressed or medically unwell have lower behavioral diversity, and as such, behavioral diversity may be an indicator of positive welfare [64]. As diversity can include behaviors associated with both positive and negative emotional valence [94], an increase in behavioral repertoire only points to an improvement in welfare if new responses lead to greater contact with positive reinforcement. For example, dogs in a kennel environment displayed more behavioral diversity compared to dogs in a home environment, and spent more time standing and locomoting, and less time lying down [142]. However, kenneled dogs also spent less time resting, and five of the 29 dogs spent more time panting, which indicates an increase in distress and a reduction in welfare [142]. While behavioral diversity reduces the likelihood of behavioral restriction and increases the likelihood that we are meeting the behavioral needs of that individual, the type of responses and their consequences must be considered. When animals have an inability to engage in behaviors that they are motivated to perform, there is a welfare compromise. Restriction of highly valued behaviors contributes to behavioral and physiological manifestations of stress, as was demonstrated by Glavin et al., [143] who found that movement restriction (response blocking) caused increased heart rate in rats. Miller et al. [64] suggest considering “appropriateness” of responses and propose that husbandry programs be developed around outcome-based metrics that lead to experiences that are meaningful to that species. To do so, they suggest (a) identifying and outlining species-specific behaviors; (b) identifying the stimuli and environment necessary to elicit or evoke species-specific responses, as well as completing a task analysis to identify response components; (c) list the adaptations that allow the animal to execute the behavior; (d) determine behavioral outcomes and how they will be measured; and (e) identify the practice, structure, or techniques that will be used to achieve behavioral goals. This leads animals to have a complete experience relative to their adaptations, increases the opportunity for animal-driven choices, and reduces the need for inputs or manipulanda provided by caretakers as supplemental enrichment [64,126].

Animals have been shown to prefer free choice over forced choice, which may be due to free choice improving survival odds in ancestors (phylogeny) or individuals learning that preferred activities and items are more often available in free choice conditions (ontogeny) [144]. The environment should be designed to induce behaviors associated with good welfare, where sensory stimuli exist on a temporal schedule that reflects what the animal would experience in their natural setting and variability promotes positive experiences. For example, foraging behaviors change throughout the year based on the season and availability of different types of food, number of resource sites, types and amounts of food available at various resources sites, distance between resources sites, and competition for resources [126,145]. While it is easy for caretakers to vary the amounts and types of foods available, it is more challenging to determine how to set the environment to promote a wide variety of foraging behaviors and ensure that the foraging setup is appropriately challenging so that the activity is fun, not frustrating. Using two Treat & Train^®^ [146,147] food delivery apparatus which functioned as resources sites, Salzer and Reed [145] examined the foraging behaviors of companion dogs by using a free operant arrangement in a daycare setting, changing the variable–time schedules of food delivery. They found that the dogs distributed themselves to maximize the delivered resources. However, given that the food was delivered on a variable interval schedule that was not also contingent on foraging behaviors, and the delivery apparatus were freely visible in a barren daycare setting, conclusions are limited. The utilization of a compound schedule of reinforcement, combining both responses and time, where responses targeted are foraging behaviors or analogous operants for foraging behaviors, would allow more information about how to use this system in a home environment to provide for more varied experiences in a restricted setting.

### 2.3. Genuine Choice

Neither freedom nor coercion exist in duality, but instead lay on opposite ends of a spectrum. Given any environment, an individual has a limited number of response options available based on their skillset, setting events, and present contingencies in effect. In behavior analysis, freedom is the availability of alternative responses, where at least two well defined behavioral alternatives are necessary for genuine choice to exist [148,149,150]. Choice responses are a function of antecedent, behavior, consequence (the ABCs), and their histories and the ABCs of alternative available patterns of responding and their histories [148,149,150,151]. 

In an effort to ensure that captive organisms have genuine choice, all contingencies currently available must be identified and the response costs and consequential benefits of alternative behavior patterns weighed. The freedom an individual has can be measured as a degree of freedom represented by n contingencies −1 [148,149,150]. If there is only one contingency in effect, this is represented by 1 − 1 = 0, demonstrating that the individual is not free, as there is no choice available. If there are two contingencies in effect, this is represented by 2 − 1 = 1 and the individual has one choice available or has one degree of freedom. In this way, freedom lies on a spectrum, and we can make our learners “more free” by teaching additional responses and providing additional reinforcers, and setting the environment so our learners can use their skills to access reinforcement in their natural environment. Each additional behavior available to an individual increases genuine choice, or the ability to choose, without coercion, from equally possible contingencies that are simultaneously available only if those contingencies exist in the environment (for a discussion of the definitions of the terms control and freedom by Skinner, Baum, Catania, and Goldiamond that are “consistent with the epistemological assumptions of radical behaviorism”, see de Fernandes and Dittrich, 2018 [148]). Alternatively, it is possible to make our learners “less free” by providing reinforcement contingent on only a small number of responses. Worse, if a behavior that is required is one that is inherently aversive to the learner, we are requiring a forced choice behavior [152]. For example, in “nothing in life is free” [153,154], dogs are taught to perform a skill, such as a sit, to earn everything that they need, including food. In this scenario, where some trainers require the dog to perform one or two specific skills to access reinforcers necessary to maintain life, the dog is lacking genuine choice. If the dog sits naturally on their own, while having 0 or 1 degree of freedom is not ideal, teaching the dog to sit is not causing additional harm. However, if the dog never sits of their own accord due to their morphology, an injury, or other circumstance, asking the learner to sit may cause pain or other aversive conditions for that animal in addition to the reinforcement they receive. Forced-choice behaviors are necessarily coercive, and as such should receive a negative value when degrees of freedom are measured. In the instance where a dog must sit for their dinner, but sitting is painful, the dog would have −1 degrees of freedom. Unfortunately, illness and injuries happen and are not always immediately apparent. Programming for multiple response options reduces the chance that a well-meaning handler inadvertently forces their dog into this unpleasant, coercive situation. 

An individual’s degrees of freedom could be an excellent welfare metric. In circumstances where a learner is living in a restricted setting (institutionalized, kenneled, or even living in a pet home), others are responsible for teaching skills, arranging the environment, providing opportunities, establishing the conditions that make consequences critical, and ensuring contingencies that provide access to those consequences are available. When multiple contingencies are equally possible, the learner has true choice [148,150]. In a scenario where a dog is required to perform any of a wide array of skills to access reinforcers necessary to maintain life [155], choice exists. The identification of both behavioral needs and forced choice behaviors is necessary for an objective assessment of the social validity of any behavior selected for intervention; therefore, degrees of freedom should be used alongside other indices to measure welfare. 

#### 2.3.1. Choice and Interventions

Hanley et al. [156] demonstrated that given a choice, participants with severe problem behavior (self-injury, aggression) preferred participating in a functional communication training (FCT; a differential reinforcement (DR) procedure in which an individual is taught an alternative response that results in the same class of reinforcement identified as maintaining problem behavior) plus punishment treatment package over FCT or punishment alone. The authors provided a choice to participants by presenting three microswitches that when pressed, functioned as the first step in a concurrent chain leading to different treatments. This created a person-centered intervention, and allowed for the objective measurement of the social validity of the treatments themselves. In addition to participants’ preference for this treatment, the combination treatment was demonstrated to be most effective. 

Rajaraman et al. [157] replicated and extended Hanley et al. [158] by incorporating an enhanced choice model to offer participants multiple choice-making opportunities, which included (a) participating in treatment involving differential reinforcement, (b) “hanging out” with noncontingent access to putative reinforcers, or (c) leaving the therapeutic space. Participants overwhelmingly chose to participate in treatment, which was successful, demonstrating that it is possible to eliminate dangerous problem behavior that previously escalated during attempts at physical management, while providing participants with the agency and control that is necessary for good welfare. In addition, by giving participants the choice to participate, Rajaraman et al. demonstrated how social validity can be placed at the forefront of a treatment package and objectively measured. The authors implemented this enhanced choice model in both an outpatient clinic and a specialized public school, and generalized skills across teachers, classrooms, and time periods, providing further evidence that skill-based treatment provided through an enhanced choice model can produce socially meaningful behavior change. 

Finally, Ramirez [159] demonstrated how to reduce the task refusal of a beluga whale by implementing a choice protocol. The whale was previously an active participant in training sessions but was no longer performing difficult and husbandry behaviors such as presenting her tail for a blood draw. After hypothesizing that the whale lost trust in the younger trainers on their care team, Ramirez provided the whale with the opportunity to perform a low-effort behavior (target) for reinforcement at any time during a training session, even if another behavior was requested. This gave her a way to say “no” when asked to perform difficult skills, and ensured there was always an alternative behavior she could offer at any time to earn reinforcement. The whale’s degrees of freedom were increased from 0 to 1, and her refusal to do behaviors was reduced from 38% to 2%. While these three examples demonstrate how behavior intervention plans can be designed with the concept of genuine choice in mind [156,157], designing a complex environment to provide the animal with multiple biologically relevant options from which to choose [124], allowing learners to choose from multiple intervention options [156], providing an additional way for learners to earn reinforcement during interventions beyond meeting target criteria [159], and giving learners the ability to opt out of the learning environment altogether [157] are not yet common practices. 

#### 2.3.2. Control over Pain 

The need for control is biologically motivated, adaptive for psychological functioning and physical health, and imperative for survival and wellbeing [160]. Choice is necessary for an individual to have a perception of control, but there is a difference between the perception of control and making choices [160,161]. The perception of control over a stressor inhibits autonomic arousal, and a vast majority of humans view having a choice as positive [160,161]. Passivity is the default, unlearned response to a prolonged aversive event that can be overcome when a learner learns how to control the aversive event [162]. Crombez et al. [163] demonstrated that having control over pain reduces vigilance in humans. When attempts to avoid pain were blocked, the participants persisted in their avoidance attempts, tried harder, narrowed their focus of attention to avoiding pain, experienced higher fear of the impending pain stimulus, and had reduced performance on a secondary task. Researchers recommended that future studies include measurement of psychophysiological responses (skin conductance and heart rate) instead of using participants’ self-reported pain levels, which suggests that this type of research could be replicated with animals if these measures are used.

Operant learning can cause a pain modulation effect. Lee et al. [164] taught 21 human participants to select from low or high pain cues, which would be followed, respectively, by mildly or severely painful stimuli. After this training, participants who selected low pain cues expected to receive mildly painful stimuli, but instead received the high pain stimuli. Participant’s pain ratings were significantly reduced when receiving the high pain stimulation after selecting the low-pain cue, compared to receiving the high pain stimulation after selecting the high pain cue. This may explain why animals who are given a choice to participate in medical handling do so even when the handling is occasionally painful. Many repetitions of pain-free trials, in addition to the choice component of the intervention, may have a similar pain modulating effect as what Lee et al. [164] demonstrated with humans. 

#### 2.3.3. Pain during Intervention

In 2004, Schilder and van der Borg [45] examined the behavioral effects of the use of a shock collar for working guard dog training using Malinois, Malinois cross, German Shepherds, and one Rottweiler. Methods used for training were determined by the trainers, and the authors report that dogs who were not trained with shock were trained with prong collars, beatings, kicks, and choke collar corrections. The most frequent administration of shock occurred after a dog was asked to do something and before the dog had the opportunity to respond to the request. Dogs were most often shocked for not “obeying” a ‘let go’ cue, heeling ahead of the handler, biting a criminal at the wrong moment, and long duration latency when told to ‘heel’. After observing changes to body language that indicate fear and distress, the authors concluded that training itself is stressful. However, all dogs observed were trained with strategies that include the application of pain, or at the very least, discomfort. The authors examined the dogs’ responses during sessions where no shock was applied and found that training with shock leads to reduced welfare post-training. In addition, behavioral responses suggested that shocked dogs made the association between presence of their owner and shock, and as such, the welfare of shocked dogs is reduced in the presence of their owner. The natural conclusion is not that training itself is stressful, but that training with pain causes stress and reduced welfare. 

As the dogs used for this study were working dogs in formal training programs, we can assume that these dogs were bred for this work. Dogs bred for working are selected to have the morphology and temperament necessary for their work. When breeding is carried out well, working dogs are genetically predisposed to performing working tasks that they find reinforcing [165]. For example, untrained Malinois puppies bite and hold objects while they are lifted off the ground, while Labrador retriever puppies do not do this when presented with the same stimulus [166]. This natural inclination toward the tasks being trained may provide a buffer from the effects of harsh training methods, as reactions to potentially stressful events depend on their meaning for the individual [167]. It is possible, but not yet demonstrated, that when pain is applied to a learner who may not have a genetic inclination toward a behavior being trained, pain may have more severe consequences. The degree to which dogs exhibit reduced welfare is positively correlated with the level of aversive control of the operant procedure used [34]. Available information suggests that the use of aversive-based methods in training is correlated with stress-related behaviors during training, fear, aggression, elevated cortisol levels, and reduced owner interaction, all of which indicate welfare was compromised [168]. Many of the completed studies are survey-based, with populations of interest that are police and/or laboratory dogs that do not represent the spectrum of breeds and temperaments of companion animals. Most studies examined the use of a shock collar specifically [168], did not include objective descriptions or measurements of training methodology, and lacked data about procedural integrity and interobserver agreement. While the data suggest that use of a shock collar is correlated with welfare compromise, we are unable to conclude that all aversive-based methods are associated with welfare compromise. Further, while most dogs demonstrated reduced welfare after experiencing harsh training methods, some did not. Many factors may contribute to this finding, including but not limited to the dog’s breed [99,109,165,169,170], physical conformation [99], temperament [99], sex [171], age, [36], physiological measures, medical conditions including behavioral diagnosis [172], rearing and learning history [18,33,34,40,99], overall quality of life before, during, and after training, owner attachment and caregiving style [172,173,174,175], dog’s motivation to perform the skill being trained, if the skill being trained with shock ultimately allowed for greater access to reinforcement, level of shock or type of aversive, number of aversive applications, inclusion of reinforcement, training methodology (appropriate criteria setting, correct timing, etc.), and procedural integrity. In military dogs, suspicion of previous rough handling, along with less time spent with the handler, was associated with fear and aggression, while dogs that lived with their handlers were more social [174]. Further, the obedience of military dogs was greater, fewer bites to military staff were reported, and dog welfare was improved when the dogs lived with their handlers and practiced a sport. 

## 3. Welfare and Private Events

To work towards a good quality of life, it is essential to understand concepts such as “subjective experiential states” or “situation-related affective states” [3,66] which refer to private events, including emotions, which are a function of the same variables and contingencies responsible for controlling public behavior [176,177,178]. Behaviors are associated with subjective experiential states such as pain and pleasure [3,179,180]. Negative affective states include “breathlessness, thirst, pain, hunger, nausea, dizziness, debility, weakness, sickness, anxiety, fear, frustration, anger, helplessness, loneliness, and boredom”, where positive affective states include “feeling energized, engaged, affectionately sociable, maternally rewarded, nurtured, secure, protected, excitedly joyful and/or sexually gratified” (see [66], Figure 2). Positive behaviors indicate anticipation, decision making processes, problem solving, investigation, preference indications, and agency, where negative behaviors indicate anxiety, fear, pain, or boredom [94]. 

In order to evaluate dogs’ affective states, researchers explored the relationship between mood states and stimulus appraisal [181], the way that emotionality impacts information processing to create cognitive biases [182], and how cognitive bias may indicate emotional valence [183]. Cognitive bias testing was used to determine how training methods affect dogs’ optimism/pessimism [34]. Dogs demonstrate behavioral laterality in response to emotional stimuli [184,185,186,187,188]. For example, there is a relationship between high levels of canine facial asymmetries and emotional and physiological distress, with dogs displaying higher levels of facial asymmetry when approached by an unfamiliar human compared to when approached by their owner or when they are alone [189]. When reunited with their owners, dogs have been shown to move their left eyebrow [190]. When presented with their owner, an unfamiliar person, and a cat, dogs wag their tails to the right, but when presented with an unknown, dominant (as established through unspecified behavioral tests) dog, they wag their tails to the left [191]. This finding was supported by Siniscalchi et al. [190] who found that dogs respond to conspecifics differently in relation to the conspecifics wagging laterality; when seeing other dogs who wag with a left-biased tail, dogs experienced increased heart rate and displayed anxious behaviors (i.e., lowering of body posture, paw lifted). This demonstrates not only that motor lateralization bias should be considered for measuring emotional valence, but that dogs are able to detect emotion in conspecifics when lateralization is present. 

While affective states may be considered constructs when viewed through the lens of applied behavior analysis, they are helpful concepts from which operational definitions can be created for individual assessments of captive animals, including companion animals. The topography of a response differs based on the context, which informs how an organism is feeling; lying down on the couch looks different than holding a down stay on a platform during an agility trial. The former is looser, with relaxed musculature, different ear and tail positions, and a larger percentage of the body in contact with the resting surface. The latter is tighter, with eyes fixed on the handler and all four limbs tucked underneath the body, as well as muscle tension sufficient to quickly launch the dog into the next response in the chain. Similarly, a dog who is comfortable having a picnic in the park with their owner might lay down on the grass in a relaxed manner as described above, where a dog who is uncomfortable in that environment would hold their body upright, primed for action should a threat appear. Further analysis into the dog’s facial expression can help the delineation of arousal from emotional valence [108,189,192,193,194]; in an agility context, a dog may be aroused and excited where in the park they are aroused and worried, as is elucidated by eye shape (round, almond, and squinting), pupil dilation, sclera visibility, the presence or absence of creases in facial muscles, tension in and position of the commissure and ears, or changes to any of the above. A two-dimensional model can be used to describe the affective state of animals by differentiating between emotional valence and arousal. Arousal can range from low to high, and valence ranges from positive to negative. 

There is a reciprocal relationship between the identification of affective states and the identification of contingencies, where the identification of one allows for the inference of the other [176]. Layng [176] proposed that private emotions are contingency indicators or descriptors which function as tacts of consequential contingencies, providing insight into their identification. Further, he posits that Panksepp’s seven types of emotions [195] describe different patterns of reinforcement. Seeking is associated with nearing an occasion for reinforcement, anger/rage is associated with removal of the other, fear is associated with the removal of oneself or the other, lust is associated with nearing an occasion for a sexual encounter, care is associated with removing distress signals, panic and grief are associated with nonspecific distancing, and play is associated with nearing reciprocal social or activity related consequences. The subcortical brain regions that “light up” during imaging when a human is experiencing an emotion exist in all mammals [196], suggesting that these brain regions, and subsequently emotional states, are homologous across species. Emotions occur with and are determined by the contingency and are accompanied by autonomic, physiological changes, which may be necessary to meet the requirements of that contingency to aid survival [176]. Behaviors start as operants, and when those behaviors contribute to survival and reproduction, the morphological and physiological structures necessary for the behavior are inherited by offspring [176,197]. Potentiating variables may not elicit respondent behavior associated with emotion, but instead might evoke canalized operants that produced reinforcement in the past [176]. These concepts echo the idea that feelings cannot be separated from other biological mechanisms when individuals are trying to cope with their environment [95]. 

Both contingency analysis and the “Five Domains” model provide frameworks for the assessment of environmental variables that influence mental state. Welfare scientists now tackle the challenge of how to measure and quantify an affective state (or covert responses and private events; [198,199]), an ability that Skinner predicted [200]. Many researchers looked at the correlations between physiological stress responses and the overt behavior of dogs during various conditions to infer affective state. Physiological, endocrine, and neural measures included immunological markers, salivary cortisol, urinary oxytocin, heart rate (HR), heart rate variability (HRV), body temperature, and respiration rate. These biological measures were compared to both dog body language (a dog’s verbal behavior) and responses such as approach and avoidance to determine if biological and behavioral responses consistently correlate with one another or are, on their own, accurate measures of the third variable, the animal’s affective state. 

### 3.1. Canine Verbal Behavior and Emotional Valence

It is necessary to determine the extent to which an animal’s behavior consistently conveys emotional valence both between and within subjects across conditions and time. Given that all responses, once elicited, contact the environment, it is critical to determine if responses exist that are not affected by environmental consequences and whose appearance always elucidates an underlying emotional response. Behaviors linked to physiological indicators of acute stress include startling, lethargy/decrease in activity levels, increase in activity levels, loss of appetite, reduced playing, hunched posture, low tail position, paw lifting, yawning, ears held low, trembling, snout/lip licking, lowered body positions, vocalizing, panting, increased salivation, repetitive behaviors, increased activity, ‘nosing’, increased urination, decreased drinking, increased salivation, increased vigilance, reduced responsiveness to humans or previous reinforcers, increased self-grooming, self-mutilation, coprophagy, hiding, destruction, change in elimination patterns, aggression, repetitive behaviors, and a change in the rate or frequency of these behaviors, in particular a lower threshold for showing any of these behaviors over time, which indicates chronic stress [99,201]. 

In reaction to receiving a shock, dogs demonstrated the following behaviors (in order of decreasing response frequency): lowering their ears, emitting a high-pitched yelp, flicking their tongues, lowering their tails, lifting their front paws, squealing, exhibiting “characteristic head movement” and avoidance behaviors, scream barking, crouching, snapping at their owners, or exhibiting no reaction [45]. After subjecting dogs to six different aversive stimuli, including sound blasts, short electric shocks, and a falling bag while measuring heart rate, salivary cortisol, and behavioral responses, Beerda et al. [201] found that with the exception of very low body posture, which was correlated with elevated salivary cortisol, the correlation between behavioral and physiological stress parameters were not significant. When the aversive stimuli included the presence of the experimenter (i.e., the dog is pushed down and held for 20 s), high levels of oral behaviors were observed. The authors indicate that oral behaviors functioned as a signal of submission or appeasement rather than a signal of stress; however, it is not clear how these constructs could be abstracted from one another. 

Camps et al. [172] retroactively examined 12 clinical cases to examine the features of pain-related aggression. Owners reported that dogs who display aggression after becoming painful exhibit a reduction or complete lack of warning prior to an aggressive display (labeled “impulsiveness”) compared to dogs who display aggression before or without becoming painful [172]. No relationship was found between cause of pain and the topography of aggressive behavior; however, dogs who were not aggressive prior to their painful condition showed aggression as a result of manipulation context more frequently. While [172] reported that pain-related aggression is a primary problem in only 2–3% of dogs who are referred to a behavioral specialist, Dinwoodie et al. [43] examined 963 dogs whose owner reported at least one aggressive response and reported 15% of dogs had an underlying medical problem. After reviewing 100 caseloads, Mills et al. [202] state a conservative estimate of behavior cases that involve pain to be 33–80% and postulate an under-reporting of the ways pain can be associated with problem behavior.

Appeasement signals are incompatible with aggressive behavior and are “context-and response-dependent sequences which are part of a ladder leading to threats or overt aggression when ignored” [203,204]. Lip licking and gaze aversion are two intraspecific appeasement signals that may also serve as an appeasement signal in dog–human communication [203]. While the term “appeasement” is mentalistic, these signals happen frequently in threatening and conflict-ridden situations [203,205], and when negatively reinforced, escalation to more dangerous distance-increasing responses are not necessary and do not occur. When not reinforced, these signals may be followed by overt aggression, including snapping and biting [205,206]. The most common behavioral changes observed before a bite include holding the body low with ears in a non-neutral position, head turning, and panting, with stiffening, staring, frowning, and snapping occurring closest to the bite [206]. Head turning, staring, and snapping decrease in frequency, plateau, or fluctuate directly before a bite occurs, with growling and “restrained” behavior observed immediately preceding a bite. Camps et al. [172] reported that dogs who displayed aggression after acquiring a painful condition were more “impulsive”, meaning they displayed minimal or no warning signals prior to an attack. 

In situations that experimenters assume to be exciting (successful problem-solving results in reward delivery), dogs demonstrated increased tail wagging and overall activity; in situations experimenters assume to be frustrating (reward delivery is unpredictable and independent of a dog’s behavior), dogs chewed the operant device available to them [102]. The relationship between consequence and behavior was consistent across reward types, which included food, social contact with a familiar human, and social contact with other dogs. The mean latency to enter the training area was negatively correlated with mean tail wags; if the dogs moved quickly to begin their task, their tails wagged more frequently. Tail wagging was historically viewed as an indicator of arousal, while tail position, which was not recorded, was historically viewed as an indicator of emotional valence [45,189,190,207,208]. The number of tail wags per second, influenced by expected reward type, was correlated with positive affective state. A less mentalistic interpretation is that tail wagging rate may be correlated with the positive value of an expected consequence. No condition included a predictable aversive consequence, so an inverse relationship between wagging rate and the negative value of an expected consequence is yet to be demonstrated, and tail wagging lateralization, which was found to correlate with emotional valence [189,190,191], was not reported. No correlation was observed between reward type and chewing frequency when the reward delivery was unpredictable. In a similar study examining facial expression, Bremhorst et al. [194] found that ear adduction was more common when a high-value food reward was delivered 5 s after the food delivery apparatus was approached, while blinking, lips parting, jaw dropping, licking nose, and flattening ears were more common when the food was withheld for 55 s. While flat ears were previously shown to signal fear, within the context of food being withheld, they may also signal frustration.

In an aim to assess a reliable measurement of fear responses in pet dogs, Ogata et al. [209] measured behavioral changes, heart rate, and body temperature during a respondent aversive conditioning procedure and found that both a remote-controlled spray collar and the conditioned aversive stimulus (a buzzer) induced significant heart rate and body temperature increases. In the first minute after the spray was presented, dogs exhibited lowered tails, running, whining, freezing, looking towards the sound, oral behaviors, panting, and jumping. Dogs in the treatment group continued exhibiting these behaviors when the buzzer was presented alone. Importantly, each dog displayed different behavioral responses both across and within subjects (see [209], Table 1), demonstrating that dogs’ behavioral responses cannot be accurately assessed at the group level. As found in previous work, behavior changes were not consistently correlated with physiological responses and autonomic changes after conditioning were found in dogs whose behavior remained consistent, indicating that autonomic reactions can be assessed more objectively than behavioral ones. 

#### Topographical Interpretation

The noninvasive measurement of animal emotions should be standard practice for the continual monitoring of an animal’s welfare quality. The most simple, noninvasive, and frequently used metric of an animal’s emotion is their facial expression and body language. After hypothesizing that emotional identification accuracy would be lowest for the Doberman, a breed with darker coloration, Bloom et al. [193] exposed Malinois, Doberman, and Rhodesian Ridgebacks to contexts that would elicit happiness, sadness, anger, fear, and disgust and took their photographs. When viewing the photos, participants were able to successfully identify these emotions at a rate significantly higher than chance. The lowest mean correct responses belonged to the Rhodesian Ridgeback, the only dog included with floppy ears, which may alter the appearance of threat. When viewing photos of dogs who did or did not receive artificial tears, participants assigned more positive scores to the photos with artificial tears [210]. Together with the finding that dogs’ tear volume increases both when they are reunited with their owners and when an oxytocin solution is applied to dogs’ eyes, these results suggest that emotion-elicited tears can facilitate human–dog emotional connections [210]. 

A qualitative behavior assessment (QBA) was developed to identify an animal’s emotional expressiveness based on the assumption that an animal’s behavior is dynamic and psychologically expressive. Adverbs are assigned to behaviors; these become a generated list of terms used for scoring observed responses. When multiple rather than singular environments are used for a QBA, the terms used to describe the animal’s emotions diversify, suggesting that QBA is sensitive to an animal’s circumstances and capable of capturing a wide repertoire of emotions [211]. This lends credence to Layng’s [176] assertion that humans know that emotional state is related to environmental circumstance, even when identifying the emotions of another species. The differences observed between shelter and owned dogs outline some of the ways in which welfare compromise is observed in shelter environments [211,212,213]. Pet dogs were labeled as more relaxed both in home and novel environments; they rest more compared to shelter dogs and exhibit fewer lip licks. Behaviors associated with increased stress (i.e., paw lifting, “displacement behaviors” such as digging/drinking, vocalization) occur more frequently in shelter compared to home environments, with dogs that remained in the shelter for more than 30 days labeled as more cautious [213]. 

Terms generated in QBAs have broad dimensions similar to each other, and are semantically consistent, with the exception of the types and frequency of terms used to describe sociability, fearfulness, and boredom [211]. Arena et al. [212] found that when terms generated by laypeople are modified by experts (in the fields of dog personality, behavior, welfare, and QBA methodology), and tested for interobserver reliability (labeled “mean clip score”) before being finalized for use, laypeople observers needed training to increase interobserver agreement. Expert definitions were not consistently operationalized and did not describe topography, but were “a brief depiction containing both qualitative and quantitative elements that together should illustrate the meaning of the term” (see [212], Table 4). This suggests that experts’ definitions of terms may differ from layperson understanding, and while terms that laypeople generate are consistent, they may not be accurate. This comes as no surprise, given our human propensity for anthropomorphism. For example, Horowitz [214] demonstrated that the body language and facial expressions that owners typically interpret as “guilt” are actually the dog’s response to being scolded independent of their engagement in disallowed behaviors. If training for IOA is included, QBA can function as a valid welfare assessment protocol. 

Inquiries about their pets’ responses land with the professionals that pet owners have most frequent contact with: veterinarians. Unfortunately, Dawson et al. [215] demonstrated that 50% of veterinarians were not able to sufficiently identify aggression in either dogs or cats, where aggression is an intentional, overt, and potentially expensive distance-increasing behavior signifying an animal is highly motivated to avoid a stimulus and is not a subtle form of communication [89]. It follows that anyone unable to identify overt aggression would also fail to identify more subtle signs of communication associated with fear, anxiety, and stress. Most veterinary colleges continue to lack veterinary behavior and applied animal welfare programs, and as such, most veterinarians lack competence in these areas and hold beliefs that do not keep pace with published data on topics including behavioral medicine and intervention [216,217]. After interviewing the staff at thirty vet clinics about their behavioral welfare practices, Dawson et al. [215] examined videos of client–staff interactions and found discrepancies between interview response and veterinary staff–patient interactions, suggesting that veterinary staff are not implementing low stress handling practices, even when they are committed to and think they are doing so. While online training programs such as Fear Free^Ⓡ^ attempt to fill the gap, not all veterinarians have exposure to this continuing education, and further, the most important component to skill acquisition, the chance to rehearse and receive feedback [218], is not provided via the online medium. 

In C-BARQ questionnaires, 41% of owners reported that their dog displayed fearful behavior while at the vet, with 14% of owners labeling their dog’s fear as extreme [219]. Dogs who are fearful are experiencing poor welfare. Many dogs are experiencing welfare compromise during veterinary visits, with potential for further welfare compromise when access to veterinary care is restricted due to the animal’s behavior while at the vet [220,221]. The Fear Free^Ⓡ^ movement campaigned to educate veterinarians about best practice to reduce fear in the veterinary setting, where recommendations include creating a low-stress environment by taking steps to reduce visual access to other patients in the waiting area, installing sound-absorbing tiles, rubberizing floors, providing separate areas for dogs and cats, using non-threatening body language, using a touch gradient when handling patients, paying attention to and adjusting the care plan based on an animal’s body language, distracting with high-value food, using prophylactic medications when necessary to reduce pain, anxiety, and fear, and teaching animals to accept veterinary procedures (cooperative care) [220,222]. Unfortunately, learning about techniques to reduce fear does not necessarily translate to reducing fear in practice, as time constraints, owner non-compliance, and space limitations may impede veterinary staff’s ability to carry out practices recommended for improving their client’s welfare while at the clinic [215,223]. 

### 3.2. Cooperative Care 

Cooperative care refers to teaching animals new skills for medical care and husbandry by giving the learner the option to leave, practicing skills frequently, and teaching the learner to accept variety. This husbandry training was pioneered by the students of B.F. Skinner (for a review, see [224]). Skilled practitioners recommend practicing painful procedures without the painful stimuli at least 100 times per each painful experience [225]. The recommendations of Ramirez [225] describe what behavior analysts would consider best practice by using positive reinforcement, skill building, and programming for generalization and maintenance. Inspired by this work, animal trainers used similar techniques with dogs. Patel [226] demonstrates how stationing on a mat and targeting a bucket with eye contact can be used to teach dogs to “opt in” to husbandry and medical handling. Bertillson and Johnson Vegh [227] developed multiple ways to incorporate “start button” and “stop button” behaviors with companion animals, where a start button behavior serves as a discriminative stimulus (or conditional discriminative stimulus) for the handler to start a procedure, and a “stop button” behavior is a discriminative stimulus for the handler to terminate the procedure. Whether labeled “The Bucket Game”, “Start Button Behaviors”, or simply cooperative care, these techniques aim to give the learner choice and control. 

While cooperative care training is becoming popular in the companion animal training sphere, there is still little empirical evidence for its use, as the effects of specific interventions on companion animal behavior in a veterinary setting were examined only twice [228,229]. Stellato et al. [228] evaluated if the implementation of a desensitization and counterconditioning program would reduce pre-existing veterinary fear in companion dogs. The treatment, which was designed by veterinarians, reflected the type of advice a general practice veterinarian might give to the owner of a fearful patient; the dog owners received only written and video instructions without an opportunity to practice or receive feedback. Treatment included gradually exposing their dog to handling of different body areas, beginning with least-invasive touch (i.e., “place hand beside their paw on the ground”) and advancing to the next level of handling when the dog was “calm” and not displaying signs of fear or discomfort (i.e., “touch their paw for 2–3 s.”). Desensitization steps did not include picking up the dog to place them on a table, examining them on a raised surface, examining with a stethoscope or otoscope, or inserting a thermometer into the dog’s anus; the most invasive type of touch that owners were instructed to use included moving hands around the body in a massage-like circular motion, which is not the type of handling that occurred during the veterinary exam. After four weeks, a small effect was shown for fear reduction in the treatment group; however, owner compliance to the intervention protocols was poor (suggesting poor social validity) and the intervention did not influence temperature, heart rate, respiratory rate, trembling, vocalizations, or the amount of encouragement needed before a dog would step on the scale. Given that desensitization to veterinary handling was incomplete, nor did it meet best practice for cooperative care training, we suggest that larger effects would be demonstrated after the completion of a well-designed desensitization program. Veterinarians are likely to engender greater success for their clients by working in conjunction with or referring their clients to animal trainers and animal behavior consultants who specialize in applied work. 

Wess et al. [229] taught dogs without a history of severe aggression toward veterinary staff (i.e., snapping, biting) to place both front paws onto and stand on a target; this was their “cooperation signal” (i.e., “start button” behavior, or discriminative stimulus for the handler to start the exam). When the dogs assumed this position, desensitization to veterinary handling began. If the dog stayed on target, they received treats. If they stepped off their target, both the examination and treats were terminated. A certified dog trainer coached participants and assessed dogs after 9–12 weeks of training and before the second veterinary visit, with results indicating that 77% of dogs exhibited moderate to very good training progress. This training was more comprehensive than Stellato et al. [228], with desensitization procedures listed commensurate with a full veterinary examination. However, the number of desensitization steps completed was not reported, and training did not include generalization to the veterinary clinic, restraint, or handling by an unfamiliar person. During testing, owners were bystanders only allowed to give their dogs treats at specified times. Even with these limitations, a stronger reduction in the mean HR between visits was related to improved tolerance of handling, and dogs whose owners performed the restraint during the exam had lower HR compared to dogs who were restrained by an unfamiliar assistant. Dogs with more pronounced trainer-rated improvement in tolerance of handling had reduced HR during visit two, suggesting that training success was correlated with lower autonomic arousal during handling. 

This study provides the first data available that may demonstrate a correlation between a reinforcement schedule (extinction) and a biological measure of emotional valence (HRV). Only one owner asked the veterinarian to stop when the dog stepped off target; for 21 of the 22 dogs, their “stop button” behavior was inadvertently placed on extinction during testing and “struggling” and “attempting to jump off the table” was reported. The authors suggest that the data indicate training had different effects on the dogs based on their previous tolerance to handling and training progress; however, given that transfer of training skills was poor, conclusions cannot be drawn between treatment success and any other measures. Additionally, latent variables not examined include the number of desensitization steps successfully completed, treatment integrity, training frequency, and consistency of stopping the exam when the dog was off target. 

### 3.3. Physiological Measures and Emotional Valence

HR and HRV are commonly used metrics that open a window to affective state by assessing autonomic nervous system (ANS) activation and could offer tools for assessing the emotional state of animals [62]. Heart rate (HR) measures the number of times the heart beats per minute, while heart rate variability (HRV) measures the vagally mediated beat-to-beat changes in heart rate. Optimal regulation occurs when autonomic arousal matches performance requirements [230]. Changes to heart rate provide information about how much an individual is having to cope with a situation, with freeze responses correlated with low heart rate and fight or flight responses correlated with higher heart rate [70]. Arousal can be measured by HR, where affective state can be measured by HRV [229]. Higher HRV is correlated with more optimal autonomic, behavioral, and emotional regulation and lower HRV is correlated with poor autonomic, behavioral, and emotional regulation and poor coping responses [229,231,232,233]. Dogs with a history of conspecific aggression, owner-directed aggression, and bite histories have significantly lower baseline HRV [231,232,233]. Low baseline HRV is additionally correlated with increased stress and anxiety [231], suggesting that this type of objective measurement might be explored as an index of welfare. After measuring changes to both HR and behaviors during and after a respondent aversive conditioning procedure, Ogata et al. [209] found that heart rate rose consistently in response to both the unconditioned and a conditioned fear-eliciting stimuli, while behavioral responding did not change reliably either between or within subjects. This suggests that measures of physiological responses could provide more accurate insight into an animal’s emotional state above observing changes in public responding.

Brugarolas et al. [234] used ECG and an electronic stethoscope to monitor HR and HRV during scent detection tasks. They observed that the highest instantaneous heart rate occurred at the beginning and end of each search, which indicates an increase in arousal. While the interpretation of patterns found were outside the scope of the study, the authors demonstrated that HR and HRV can be obtained during intervention to be interpreted as a welfare metric. Polar^®^ human heart rate monitors were validated for use in dogs [229,231,235], allowing for an accessible way for pet owners, shelter staff, trainers, or veterinarians to measure HR and HRV, with alternate tools of measurement for use in pets becoming commercially available (i.e., PetPace collars or similar purchasable pet monitoring products).

Significantly increased HR in dogs is correlated to types of animal-assisted interventions (AAI) and duration of travel [236], an increase in the appeasement gestures ‘tongue flick’ and ‘paw lift’ [237], barking and growling at a stranger [231], threatening encounters [106], aversive conditions including unexpected sound blasts, unexpected short electric shocks, unexpected falling bags, opening umbrellas, restraint [201], and a conditioned stimulus ‘sound buzzer’ after being conditioned with an unconditioned aversive stimulus, a spray collar [209]. 

Zupan et al. [238] demonstrated that exposure to stimuli confirmed to be positive resulted in changes to nine research beagles’ HRV parameters, suggesting that higher positive emotional valence in dogs is associated with parasympathetic deactivation. Katayama et al. [239] collected data via ECG and found that HRV changed when dogs were exposed to an appetitive condition (owner petting) and an aversive condition (owner leaving the dog alone in a novel space); the type of change observed could be used to determine the dog’s emotional response. Similarly, Gácsi et al. [106] found that dogs who made distress vocalizations during separation from their owners and who growled or barked at a threatening stranger (dogs labeled “reactive”) experienced a decrease in HRV during the threatening encounter. Together, this suggests that measuring HRV during behavior interventions could determine how the animal feels about that intervention. As interpretations of HRV measurements and their implications are not yet universal (for a similar review of HRV research that reaches different conclusions, see Polgár et al., 2019 [183]), future research on baseline HRV measures and HRV changes in dogs will allow us to better understand the suitability of these measures as an applied welfare metric during intervention, especially as developing technologies for the measurement of HR and HRV become increasingly available. We did not include all available measures in this review, as many are impractical for applied use, as they cannot be easily obtained nor immediately interpreted. For example, the relationship between cortisol:creatinine ratios (C/Cr) and stress were examined, and even though a negative correlation between C/Cr ratio and lip licking was discovered, the validity of this measure was questioned [142]. Cortisol is typically measured via saliva, urine, or hair; while taking samples and waiting for results may be a useful tool for a veterinarian measuring stress over time [240] and should be considered when available, it may be cost prohibitive and fails to immediately inform the applied practitioner about the animal’s welfare during the intervention underway. See Csoltova and Mehinagic [241] for a review of all available neurobiological and psychophysiological measures for the assessment of positive dog emotion.

## 4. Discussion

Aforetime, both ethologists and radical behaviorists focused on directly observable (overt) responses [200,242]. This set a precedent that persists to this day, with few studies measuring internal events that eluded direct measurement in the past. With the advancement of technology, we are now able to combine behavioral, physiological, and biochemical measures, including endocrine and neural measures, for the consideration of a learner’s emotional affect as one measure of their welfare [3,140,241]. 

Ogata et al. [209], Fraser [59], and Gácsi et al. [106] note that dogs’ behavioral responses and physiological reactions to the environment are individual and may not be accurately assessed at the group level, yet single subject design is not prolific among published welfare or training research. As behavior is an epiphenomenon that does not cause or explain the occurrence of behavior [58], single subject design [243,244,245] provides methods for isolating the environmental variables controlling a learner’s response, affect, and ultimately their welfare and quality of life. 

A complete understanding of the biological basis for stress, [220] ethological, niche related mechanisms [246], and group research that elucidates most animals needs and preferences [58,59] provides a robust framework from which to design habitats that maximize antecedent arrangements that elicit variable species-typical behaviors in order to improve animal welfare. This knowledge is necessary for creating hypotheses about the setting events, antecedents, and consequences responsible for individual behaviors that may need intervention and which can be tested with within-subject methodology. The development of new technologies allows us to look “under the skin” and account for covert responses that can now be observed objectively. In combination with single-subject design that allows for the measurement of individual behavior change, welfare metrics must be used to assess how a learner feels about the intervention underway and to determine how applied interventions affect the learner’s quality of life. To this end, we make the following recommendations:
(1)Target responses selected for intervention should meet a behavioral need, or replace a forced-choice behavior, and increase the learner’s degrees of freedom. This prompts practitioners to carefully consider if they are using coercion to achieve a goal, even when positive reinforcement is programmed as reinforcement; (2)Operational definitions of target responses should include qualitative descriptions of responses that indicate positive affective state. This highlights an additional way to gauge how a learner is experiencing a contingency in effect in order to promote a positive learning experience. For example, while a learner may ultimately express frustration by mouthing on an object, as was observed by McGowan et al. [102], being aware of and looking for a more subtle change, such as the flattening of ears [194], would allow practitioners to stop and modify the intervention before frustration, and the responses that accompany it, which are not part of the target response, escalate;(3)Practitioners should improve the chance that an intervention is socially valid to the learner by ensuring that interventions (a) stem from educated hypotheses created based on both population and individual data, taking the learner’s adaptations, perception, and cognition into account, (b) use functional reinforcers evaluated by functional analysis and/or preference assessments, rather than presumed or contrived reinforcers, (c) program for choice, allowing the learner to choose to participate, (d) provide genuine choice within the intervention by providing the learner the opportunity to perform an alternate, non-target behavior to earn the same reinforcer available for performance of the target behavior, (e) improve the learner’s relationship to, or teach additional coping skills that can be used when the learner must come into contact with an unavoidable aversive stimuli that exists, and will exist, in their environment, where appropriate, and (f) program for contact with positive reinforcement, even when positive reinforcement is not used at the outset of an intervention. These recommendations set the occasion for learner-centered interventions and increase the likelihood that welfare is positive, or at least improved, in learning contexts;(4)To ensure that the intervention is having a positive impact on welfare, quality of life metrics that can be repeated across time should be completed before, during, and after the intervention, and affective state should be measured during the intervention using validated metrics, such as body language, HR, and HRV.

While this paper focused on the behaviors of, interventions for, and welfare concerns of companion animals, namely pet dogs, the outlined recommendations may be applied to any learner. Where artificial conditions are inevitable for animals in captivity, humans who are neurodivergent also find themselves in suboptimal environments that disallow for their needs. Everything from lighting to social expectations are designed for neurotypical individuals. The responses used for managing overstimulation caused by the environment are considered “inappropriate” and often socially punished (for example, hand flapping) [247,248], while responses which may be forced-choice behaviors (e.g., eye contact; [249]) are generally promoted. For these reasons, some neurodivergent individuals may be prohibited from experiencing optimal welfare in public settings [247,250,251,252]. Behavior analysts “serve the status quo” [60] by promoting neurotypical behavior [249] that may actively cause harm. 

A lack of social validity from the client led to public outcries against ABA services, with some autistic adults (who received behavior analytic services as children) sharing that their experiences with ABA were abusive and led to trauma [253,254,255]. While these claims warrant further investigation [256], and it may be the quality of behavior analytic services implemented and not the technology itself that caused harm, if behavior analysts are going to take appropriate steps to “support clients’ rights, maximize benefits, and do no harm” [257], include ethics-based criteria for implementing treatments, and adopt a trauma-informed care model [258], the acceptability of target behaviors and their interventions must be centered around the interests of the clients themselves. To accomplish this, practitioners must have a comprehensive understanding of the cognitive abilities, perspective, and needs of the individuals with which they work.

## 5. Conclusions

We believe the recommendations outlined in this paper provide a guide for increasing the ethicality of behavior analytic interventions, no matter who the learner is that is being supported. These recommendations provide multiple technologies for the objective measurement of how a learner values various responses and resources, allows for more accurate inference to a learner’s affective state, and promotes the adoption of interventions that provide choice to the learner. Taken together, scientists and practitioners gain robust metrics from which social validity can be more accurately inferred. This is an important factor that cannot be underrepresented in the provision of ethical services [150,256,259] in order to promote a good quality of life.

## Data Availability

Not applicable.

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
