# Peer review of "The Science and Social Validity of Companion Animal Welfare: Functionally Defined Parameters in a Multidisciplinary Field"

_animals, 2023, doi:10.3390/ani13111850_

Round 1
Reviewer 1 Report
I greatly enjoyed reading this paper. It provides a comprehensive review of the currently available literature from multiple scientific disciplines with regards to the effects of human interventions (in the broadest term of the word) on animal welfare, and how we can use the knowledge and technology available to us to ensure an optimal welfare outcome for animals that are undergoing interventions.
I found the paper covered the topic it was reviewing very comprehensively, and in a clear and easy-to-follow manner. The recommendations that the authors make at the end of the paper follow logically from the review and discussion of the topics that were covered in the main body of text. In my opinion this is a timely review, and a good overview of ways in which the welfare outcomes of animal interventions can be improved. It adds good information to the literature, and will be a great help for researchers or practitioners, providing them with tools and ideas on how to ensure the best outcomes for their animal subjects. It is also written in clear and well-formulated English, which I could not detect any issues with other than perhaps a typo or two.
I believe this paper to be of high quality, and I cannot make any recommendations on how to further improve it.
Author Response
Hello Reviewer 1,
Thank you so much for taking the time to read my lengthy manuscript, and for the generous feedback. This manuscript was a labor of love and I am so happy to read your comments! I did make a few edits based on some other reviewer feedback and am hopeful that the work can be shared with others soon.
Best,
Lauren Novack
Reviewer 2 Report
The authors deserve credit for their attempt to present the complex problem of animal welfare. The possibilities of assessing the phenomenon are subjective, and the search for and improvement of forms of assessing the quality of animal life requires the accumulation of practical knowledge and professional literature.
The manuscript presents a rich literature, correctly cited, but in my opinion the notation "from – to" would improve readability, for example, [25-32] instead of [25,26,27,28,29,30,31,32];
The attractiveness of the work, and thus readability and citation, would be increased by introducing figures, charts and tables, which I encourage the authors to do, such as simplified QOL questionnaires and other mentioned welfare assessment parameters.
However, all the comments concern the aesthetics and readership attractiveness of the manuscript, not the substantive values, so the work in my opinion is suitable for printing.
Author Response
Dear Reviewer 2,
Thank you for taking the time to read my lengthy manuscript. I completely agree with your note about the citations and have made those changes.
Warmly,
Lauren Novack
Reviewer 3 Report
This is an very insightful review of the role of applied behaviour analytics in canine welfare. The review is well written overall and I only have a few comments below as suggestions.
There have been a large number of citations in the first couple of paragraphs and I wonder if these are necessary. Perhaps on page 2, paragraph 2 those citations should appear after each of the aspects of cognition is mentioned.
Glanville et al 2020 is not in formatting style
Page 3 paragraph 3 citation 2 should have author name in text?
I think there is a good summary of welfare but given the length of the paper I wonder if this could be reduced to be more concise and linked earlier to ABA as the key aims of the paper.
Page 4 paragraph 2 ethics statement is a nice point, could an example be given to go with this.
At the end of page 7 could a caveate be added to ensure consideration of the welfare of other animals when dogs are being allowed freedom off lead- in particular in relation to livestock and also wild animal welfare.
2.3.1 FCT needs to be defined for the reader
Definition of cooperative care refers to practicing painful procedures but it can also relate to uncomfortable procedures and not always pain- for example nail clipping. I would question the statement of practicing painful procedures 100 without painful stimuli, there is no science to back this up.
HR and HRV are introduced before being defined. I feel section 3.3 tries to tackle a huge area and feels a little rushed at the end. I wonder if reviewing all physiological measures is beyond the scope of the paper and HR and HRV can be introduced earlier in the previous section and the main body of the paper end on the points of linking ABA to cooperative care.
Author Response
Hello R3! Thank you so much for taking the time to provide excellent feedback on my lengthy manuscript. I have made the following changes:
- Citations on page 2 paragraph 2 have been made more concise; [4-15]
- Glanville et al 2020 is now in appropriate formatting style (great catch, thank you!)
- Page 3 paragraph 3 citation now has author name in text (I wasn't sure how Animals wanted to handle this type of in-text citation, thank you for the clarification!)
- An example has been added after the ethics statement on page 4 paragraph 2
- A caveat was added to the end of page 7 re: welfare of other animals when freedom is allowed off leash
- Definition of FCT added
- I adjusted the language when defining cooperative care to make it clear that the recommendation of practicing a painful procedure 100 times without the painful experience is a recommendation made by a skilled practitioner
- One of the authors got married in the last few months; her last name is now hyphenated
There are two recommendations made that I wholly appreciate, but would like to leave as-is.
The first: "I think there is a good summary of welfare but given the length of the paper I wonder if this could be reduced to be more concise and linked earlier to ABA as the key aims of the paper"
Thank you for the feedback. While I agree that this could and should be reduced for a welfare science-literate audience, not many applied practitioners are familiar with these concepts. In fact, in the dog training world, the 5 Freedoms are still taught at conferences and in seminars as the metric by which to assess welfare. I do feel that this brief history best makes the distinction between the 5 Freedoms and 5 Domains, and explains why using the up to date framework is important. Additionally, I love how the 5 Domains are explained through an ethological lens, which is one many dog trainers are most familiar with. The parallel to shifts in ABA is pointed out at the end of the first paragraph, which is important due to the lack of integration between these fields and global misunderstandings surrounding ABA. Given that this is an open access journal that many dog trainers do and will access, I'm hoping you agree that keeping this context in the paper as-in will be beneficial to one of our target audiences. The second: "HR and HRV are introduced before being defined. I feel section 3.3 tries to tackle a huge area and feels a little rushed at the end. I wonder if reviewing all physiological measures is beyond the scope of the paper and HR and HRV can be introduced earlier in the previous section and the main body of the paper end on the points of linking ABA to cooperative care." My aim was to focus on neurobiological measurements that had been taken with companion pet dogs, specifically as it relates to any type of intervention. Not that much literature exists, which is why it's a shorter section. In order to inspire others to include these biological measures in future research, I'd like to keep the section as is. For example, it would be wonderful to compare HR and HRV before and after various interventions for dogs diagnosed with various types of aggression and anxiety, and to use these measurements with stoic dogs to supplement body language scores as a measure of their experience of an intervention.

Reviewer 4 Report
The study of animal welfare is a current topic that stimulates scientific research to identify standardized protocols. Different aspects are taken into consideration in the study from a multidisciplinary perspective. On the basis of the pre-existing scientific literature, an articulated network of evaluations is reconstructed; the quality of life of dogs, for example, should also be considered in the interaction with its owner and in the environment in which they live; animals should be free to express behaviors so that there is a genuine choice, configuring the possibility of attributing a valid measurement of well-being. Pain assessment is also considered, both in the context of clinical practices by the veterinarian and during dog training, as it causes stress and reduces animal welfare; finally, the evaluation of animal affective states is also proposed. Overall, the study is well written and provides interesting points of view regarding the parameters to assess animal welfare.
Author Response
Hello Reviewer 4,
Thank you so much for taking the time to read my lengthy manuscript. Some minor edits have been made based on the recommendation of other reviewers. I'm hoping to be able to share this with others soon!
Warmly,
Lauren Novack